# Genome-wide Analysis of Basic Helix-Loop-Helix Family Genes and Expression Analysis in Response to Drought and Salt Stresses in *Hibiscus hamabo* Sieb. et Zucc

**DOI:** 10.3390/ijms22168748

**Published:** 2021-08-15

**Authors:** Longjie Ni, Zhiquan Wang, Zekai Fu, Dina Liu, Yunlong Yin, Huogen Li, Chunsun Gu

**Affiliations:** 1College of Forest Sciences, Nanjing Forestry University, Nanjing 210037, China; LongJieNi@njfu.edu.cn (L.N.); fzk@njfu.edu.cn (Z.F.); cc0212@njfu.edu.cn (D.L.); 2Institute of Botany, Jiangsu Province and Chinese Academy of Sciences, Nanjing 210014, China; wangzhiquan@cnbg.net (Z.W.); ylyin@cnbg.net (Y.Y.); 3Jiangsu Key Laboratory for the Research and Utilization of Plant Resources, Jiangsu Provincial Platform for Conservation and Utilization of Agricultural Germplasm, Nanjing 210014, China

**Keywords:** bHLH, molecular biology, salt and drought stress

## Abstract

The basic helix-loop-helix (bHLH) family of transcription factors is one of the most significant and biggest in plants. It is involved in the regulation of both growth and development, as well as stress response. Numerous members of the bHLH family have been found and characterized in woody plants in recent years. However, no systematic study of the *bHLH* gene family has been published for *Hibiscus hamabo* Sieb. et Zucc. In this research, we identified 162 bHLH proteins (HhbHLHs) from the genomic and transcriptomic datasets of *H. hamabo,* which were phylogenetically divided into 19 subfamilies. According to a gene structural study, the number of exon-introns in *HhbHLHs* varied between zero and seventeen. MEME research revealed that the majority of HhbHLH proteins contained three conserved motifs, 1, 4, and 5. The examination of promoter *cis*-elements revealed that the majority of *HhbHLH* genes had several *cis*-elements involved in plant growth and development and abiotic stress responses. In addition, the overexpression of *HhbHLH2* increased salt and drought stress tolerance in *Arabidopsis*.

## 1. Introduction

Transcription factors (TFs) are protein molecules with distinct structural and functional characteristics that influence gene expression. In the model plant, *Arabidopsis thaliana* around 100 TFs have been characterized, and these are crucial for the plant [1]. The basic helix–loop–helix (bHLH) transcription factor family is the second biggest protein family found in plants [2]. It is characterized by a conserved domain consisting of 40–60 amino acids, which consists of two parts: the basic region and the HLH region. The basic region is located at the N-terminus and is composed of 13–17 basic amino acids, which are used as the DNA binding region to identify and specifically bind DNA promoters and target genes. The HLH region is found at the C-terminus of the bHLH domain and consists of two parental helixes bound by relatively distributed (length and primary sequence) loop regions that promote protein interactions [3,4,5]. An increasing body of evidence indicates that *bHLH* genes are involved in metabolic regulation, plant growth, and production, and responsiveness to environmental signals in plants. The maize *R* gene, which is required for anthocyanin production, was the first member of the bHLH family to be identified [6]. Since that time, a growing number of *bHLH* genes have been found in various plants. A growing amount of evidence suggests that bHLH TFs are important for engagement in plant defense responses, particularly in response to abiotic stress responses such as drought, low temperature, high temperature, and salt stress [7,8]. For example, transferring the *MfbHLH38* gene from *Myrothamnus flabellifolia* to *Arabidopsis* greatly improved the resistance to drought and salt stresses in transgenic plants [9]. Wheat *TabHLH49* gene controls the expression of the dehydrin *WZY2* gene, improving drought tolerance [10]. Overexpression of the MYC-type *bHLH* gene *ZjICE2* in *Zoysia japonica* rice resulted in increased chlorophyll content and photosynthesis ability, as well as high cold tolerance [11]. By controlling plant stomata density, stomatal aperture, photosynthesis, and development, the *bHLH* gene *PebHLH35* in *Populus euphratica* was shown to act as a positive regulator of the drought stress response [12].

The *bHLH* gene family has been identified in several organisms so far, and has been separated into 15–26 classes in plants based on sequence similarity [13]. In *Arabidopsis thaliana*, for example, 162 *bHLH* genes were discovered and divided into 21 subfamilies [14]; 115 *bHLH* genes, divided into 21 subfamilies, were identified in spine grapes [15]. In red walnut, 102 *bHLH* genes were discovered, which were split into 15 subfamilies [16]. In pepper, 122 *bHLH* genes were discovered and split into 21 subfamilies [17]. However, no study on the *bHLH* gene family has been conducted in *Hibiscus hamabo* Sieb. et Zucc.

*H. hamabo* is a shrub plant in the genus Hibiscus, family Malvaceae, and an important semi-mangrove plant [18]. It is commonly used in forests, wetlands, and coastal beaches because of its outstanding salt resistance and fine morphological characteristics [19,20]. At the same time, *H. hamabo* is also a good material for investigating the mechanism of salt tension in woody plants. Therefore, it is very important to research the *H. hamabo bHLH* gene family to understand the semi-mangrove plant biological processes in response to stress.

In this study, bioinformatics was utilized to identify members of the bHLH family and to assess their relevant features using genomic sequence and full-length transcriptome datasets. Additionally, we screened for *bHLH* genes that may be involved in the salt and drought stress biosynthesis pathways in *H. hamabo*. The study aimed to increase our awareness of the *bHLH* functions involved in the salt and drought stress responses of *H. hamabo*, while, our comprehensive research laid the groundwork for additional mechanisms of salt and drought tolerance for *bHLH* genes in *H. hamabo*, with particular emphasis on identifying potential genes implicated in *H. hamabo* salt and drought stress responses.

## 2. Results

### 2.1. Identification and Physicochemical Properties of bHLH Proteins in H. hamabo

To find *bHLH* genes in the *H. hamabo* genome, we used a hidden Markov model (HMM) file (PF00010) to conduct a genome-wide HMM-search. There were 167 putative bHLH proteins discovered. The bHLH domains were confirmed by using Pfam online database, and redundant sequences were deleted. Eventually, we obtained 162 sequences from the *H. hamabo* bHLH family, which we named *HhbHLH1*–*HhbHLH162* based on their chromosomal position. The 162 *HhbHLHs* gene protein sequences are shown in Appendix A. To better characterize these *HhbHLHs* genes, we examined their physicochemical properties. The 162 HhbHLHs proteins differ in Coding sequence (CDS) length, protein length, molecular weight, and theoretical isoelectric point (pI) (Appendix A). Specifically, the CDS lengths ranged from 714 (*HhbHLH137*) to 2169 (*HhbHLH133*), while their pIs were between 4.35 (*HhbHlH67*) and 10.03 (*HhbHLH43*). Furthermore, the 162 *HhbHLH* genes were spread at random across 46 chromosomes. The remaining 42 chromosomes, except chromosomes 28, 30, 35, and 46, had varying numbers of *HhbHLH* genes. Among them, chromosome 24 contained the largest number (10 *HhbHLHs*) (Appendix A).

### 2.2. Multiple Sequence Alignment, Phylogenetic Analysis, and Classification of HhbHLH Genes

Multiple sequence alignment of bHLH domains was used to determine the phylogenetic relationship of HhbHLH proteins, which span across 40–50 amino acids. The bHLH domains of 19 different *Arabidopsis* bHLH proteins from each of the subgroups were chosen at random for further comparative studies. As shown in Figure 1, the sequences in the bHLH domain were strongly conserved [21], and the sequences of different subgroups were very similar in comparison with *Arabidopsis*.

We studied and drew the unrooted phylogenetic tree applying the neighbor-joining (NJ) method based on the similarities between *Arabidopsis* and *H. hamabo* bHLH domains. The result revealed that the 162 *HhbHLHs* gene can be classified into 19 distinct clades. Among them, the VIIa+b subfamily had the most members (20 *HhbHLHs*), whereas Groups Ib, II, IVc, and X subfamily had the fewest members (one *HhbHLH*). In addition, the *H. hamabo* bHLH subfamily was extended or contracted to different degrees in comparison with *Arabidopsis*. The VIIa+b subfamily of *Arabidopsis*, for example, had 15 members, which grew to 20 in *H. hamabo*. In subfamilies Ia and Iva, a similar condition existed, while in the X subfamily there was a substantial reduction in the number of *H. hamabo* genes, from nine to one in *Arabidopsis* (Figure 2).

### 2.3. Gene Structure and Motif Composition of HhbHLH

The motif type and composition can determine the function of the protein, and an assessment of its preserved motifs can also be used to determine the evolutionary relationship between the HhbHLH proteins. MEME software was used in this analysis to explore the motif of 162 HhbHLH proteins, yielding a total of 20 motif structures. Other than motifs 1, 4, and 5, which are broadly dispersed bHLH domains, HhbHLH members belonging to the same classes typically share a similar motif composition. (Figure 3b), such as, the XI, XII, and IX subfamilies, which indicates that there may be functional similarities between HhbHLH proteins of the same subfamily. Furthermore, certain motifs are only found in some subgroups, for example, motif 16 occurs only in VIIa+b, and motif 15 appears only in IVd. These conserved motifs may play an important role in specific subgroups, and their specific functions remain to be elucidated.

We analyzed the structure of all *HhbHLH* genes, as seen in Figure 3c, to gain an understanding of the bHLH family evolution in *H. hamabo*. Exon numbers ranged from 1 to 17 in 162 *HhbHLH* genes. There were no introns in 15 of the genes. These 15 intron genes were divided into two subfamilies, with 13 belonging to the IIId+e subfamily and two to the VIIb subfamily. In addition, the results revealed that the *HhbHLH*s have not only a large number of exons but also have genetic structure diversity. Exon distribution patterns in the IVd and VIIIc subgroups, for example, are relatively conservative. The number of exons, their distribution, and the length of each exon is all very similar, whereas the exon distribution in the XI and XII subgroups is not. Overall, the phylogenetic study findings, along with the motif compositions and related gene structures of the *bHLH* members of the subfamily, could support the group classifications reliability.

### 2.4. Chromosomal Distribution and Synteny Analysis of HhbHLH Genes

Gene duplication events, such as whole-genome duplication (WGD)/segmental duplication, tandem duplication, and transposition, are the primary causes of gene family expansion and have a significant effect on the evolution of gene families [22,23]. Through the use of MCScanX software to detect the duplication events of the *HhbHLH* family, 162 *HhbHLH* genes were divided into three repetitive events, of which 150 (92.59%) genes were assigned to WGD/segmental events, and 11 (0.067%) genes belonged to the dispersed type. The results indicated that the HhbHLHs gene family expanded primarily as a result of WGD/segmental events. (Table 1 and Appendix A).

To better understand the evolutionary process of the *HhbHLH* genes, we used intragenomic synteny analysis to identify conserved chromosome blocks within *H. hamabo*. We mapped 162 *HhbHLH* genes on 46 chromosomes using annotation information of the genome-wide data. Except for chromosomes 28, 30, 35, and 46, the remaining 42 chromosomes had different numbers of *HhbHLH* gene distributions, of which chromosome 24 contained the most (10 *HhbHLHs*). Furthermore, there was no positive relationship between the number of genes on a chromosome and its length (Figure 4). We used intragenomic synteny analysis to investigate the evolutionary mechanism behind the *HhbHLH* gene and discovered that there are a total of 68 pairs of *HhbHLH* genes that had a collinearity relationship (Figure 4). The Ks value can be used to estimate the evolutionary date of WGD/segmental duplication events (synonymous substitutions per site) [24]. According to the magnitude of the Ks value, we found that the *HhbHLH* gene family WGD/segmental events occurred in different periods; some occurred in a distant period (Ks > 1) and others occurred in a relatively recent period (Ks < 0) (Appendix A). The Ka/Ks ratio can be used to represent the selection intensity and direction. A Ka/Ks value of one indicates neutral evolution, a Ka/Ks value greater than one indicates positive selection, and a Ka/Ks value less than one indicates purifying selection [25]. The Ka/Ks ratio of all homologous *HhbHLH* genes was less than one in the *HhbHLH* gene family. This result suggests that *HhbHLHs* evolved primarily under purifying selection (Appendix A).

We created comparative syntenic maps between *H. hamabo* and *Arabidopsis* and *Populus* to get a better grasp of the phylogenetic mechanisms underlying the *H. hamabo* bHLH family. In total 60 *HhbHLH* genes showed a syntenic relationship with those in *Arabidopsis*, and 26 *HhbHLH* genes showed a syntenic relationship with those in *Populus* (Figure 5). Some of the *HhbHLH* genes, such as *HhbHLH11* and *HhbHLH141*, were related to at least three syntenic gene pairs (especially in *Arabidopsis*). Additionally, several *bHLH* collinear gene pairs identified between *H. hamabo* and *Arabidopsis* were linked to highly conserved syntenic blocks spanning over 100 genes. This phenomenon suggested that an evolutionary relationship may exist between the *bHLH* gene family of *H. hamabo* and that of *Arabidopsis*.

### 2.5. Promoter Analysis of HhbHLH

In this study, 162 *cis*-acting elements in the upstream 2000 bp sequence of the *HhbHLH* gene were analyzed and, finally, a mass of potential *cis*-elements was obtained (Appendix A). The visualization of nine *cis*-acting elements that are more important in stress response is shown in Figure 6. The results show that genes responsive to light, low temperature, gibberellin, MeJA, drought stress, wounds, and MYB binding sites involved in drought and other elements related to stress response were abundantly enriched in the promoter region of the *HhbHLH* gene. This result indicates that the *HhbHLH* gene may be involved in a variety of abiotic stress response and defense mechanisms.

### 2.6. Expression Profile and Patterns of HhbHLH Genes in Response to Drought and Salt Stresses

We examined the expression patterns of 162 *HhbHLH* genes using transcriptome data from *H. hamabo* leaves under drought (15% PEG6000) and salt (400 mM NaCl) stress conditions in this study. Of the 162 *HhbHLH* genes, ten genes were not expressed in the transcriptome data, while the remaining genes were expressed (FPKM > 0) (Appendix A). The expression of some genes showed the same trend under the two stresses. For example, the *HhbHLH2* of subfamily XI, *HhbHLH21* and *HhbHLH48* of subfamily IIId+e, *HhbHLH31* of subfamily II, and *HhbHLH35* genes of VIIa+b were up-regulated under both stresses. *HhbHLH31* from subfamily II, *HhbHLH75* from subfamily IIIf, and *HhbHLH78* from subfamily XII were all down-regulated under both stresses. The results suggest that these genes may play a critical role in *H. hamabo* response to drought and salt stress. Some genes have different expression patterns under the two stresses. For example, *HhbHLH20* of subfamily Ia, *HhbHLH30* of subfamily XII, and *HhbHLH67* of subfamily III by are significantly up-regulated under drought stress and down-regulated under salt stress, while the XI c subfamily, *HhbHLH83,* and *HhbHLH29*, were significantly up-regulated under salt stress and down-regulated under drought stress, indicating that these genes may have different regulatory modes in *H. hamabo* under drought and salt stresses. Furthermore, under both stresses, the genes of subfamily VIIIb and VIIIc had lower expression levels, implying that the involvement of these two subfamilies in the *H. hamabo* reaction to drought and salt stresses may not be apparent (Figure 7a).

We chose 15 *HhbHLH* genes at random to study their function further by quantitative real-time PCR (RT-qPCR). The expression trends of these genes under NaCl and PEG treatment for 0 h, 6 h, and 24 h were shown to be largely compatible with the transcriptome data, demonstrating the reliability of the transcriptome data. At the same time, three time points (1 h, 2 h, 12 h) were added to the analysis of the expression profiles of the genes under drought and salt stresses to further understand the changes of gene expression level in different treatment periods of stress. Compared with the transcriptome data, the expression trend of some genes was more abundant after the new time point was added. For example, *HhbHLH51*, *HhbHLH78*, *HhbHLH82*, and *HhbHLH84* showed a significant up-regulation after 1 h of drought stress and were then significantly down-regulated. A similar expression was observed in *HhbHLH57*, *HhbHLH78*, *HhbHLH136*, and *HhbHLH78* under salt stress. This phenomenon indicates that these genes may play different roles in different periods of stress. At the same time, some genes, such as *HhbHLH20* and *HhbHLH26*, were significantly up-regulated during drought and salt stresses after 12 h, while there was no obvious differential expression in other periods. In addition, some genes have different expression trends under drought and salt stress. For example, *HhbHLH8* is up-regulated after 1 h of drought stress, but down-regulated after 1 h of salt stress. *HhbHLH20* is up-regulated after 12 h of drought stress and down-regulated after 12 h of salt stress. These results show that under different treatment times, some genes may respond to the two stresses through different regulatory ways. In conclusion, the results of RT-qPCR effectively supplement the transcriptome data and provide a reference for the future exploration of the function of the genes (Figure 7b,c).

### 2.7. Expression Patterns of HhbHLH Genes in Response to Different Treatments

To further understand whether the above 15 *HhbHLH* members would be affected by other abiotic stresses and hormone treatments, we used RT-qPCR to analyze their expression patterns under low temperature, high temperature, abscisic acid (ABA), and salicylic acid (SA) treatments. In general, some genes with higher expression levels under drought and salt stresses showed similar expression patterns under abiotic and hormonal stress. For example, *HhbHLH2*, *HhbHLH8*, and *HhbHLH26* had similar expression patterns under the four stresses. Both had a significant trend in up-regulation. This result is similar to their expression under drought and salt stresses, indicating that these genes may play a very important role in the response of *H. hamabo* to stress. Some *HhbHLH* genes can be significantly induced or inhibited by a variety of treatments. For example, *HhbHLH2* and *HhbHLH82* have a significant response to low temperature, high temperature, ABA, and SA treatments. One treatment can also induce the expression of multiple *HhbHLH* genes at the same time. For example, cold treatment simultaneously induces the expression of *HhbHLH2/20/26/29/57/84* and other genes. In addition, some genes had the opposite expression patterns under different treatments. For example, *HhbHLH20* is significantly up-regulated under SA stress but exhibits a significant down-regulation under ABA stress. This phenomenon indicates that *HhbHLH* family genes can participate in multiple conditions, and their regulation is different under different stresses (Figure 8).

### 2.8. Overexpression of HhbHLH2 Increased Salt and Drought Stress Tolerance in Arabidopsis

Previous research has shown that the *bHLH122* gene in *Arabidopsis* can be expressed in significant amounts under drought stress and that plants with the *AtbHLH122* gene overexpressed are more drought tolerant than wild plants [26]. The *HhbHLH2* gene, which belongs to subfamily IX, showed higher expression levels in drought and salt stresses in *H. hamabo* leaves. This result prompted us to further explore its potential functions under drought and salt stresses. To study the function of the *HhbHLH2* gene, which is driven by CaMV 35S promoter this was transferred into *Arabidopsis* (Col-0) via Agrobacterium-mediated transformation, and homozygous T3 generation plants (OE-3, OE-5, and OE-6) were obtained through screening. Under normal conditions, there was no obvious growth difference between wild-type and transgenic *Arabidopsis* (Figure 9a,b). As the NaCl and D-mannitol(D-M) concentration continued to increase, although the root length of WT and transgenic plants decreased consistently, the decline in transgenic plants was slower than that in WT plants. Under 50 mM NaCl, the root length of WT plants was reduced by 57.98%, while the root length of transgenic plants was reduced by 11.64%–53.54%. The average root length of the transgenic plants was 1.72-fold that of the WT (Figure 9c,d). Under 200 mM D-M, the root length of WT plants was reduced by 80.88%, while the root length of transgenic plants was reduced by 2.28–12.00%. The average root length of the transgenic plants was 5.02-fold that of the WT (Figure 9e,f). Overall, these results indicate that *HhbHLH2* can participate in the regulation of drought and salt stress responses, while overexpression of *HhbHLH2* can endow transgenic plants with drought and salt stress tolerance.

## 3. Discussion

The bHLH protein family is one of the largest TF families in plants and can participate in many pathways such as plant growth and metabolism. The current *bHLH* gene family genome-wide identification work has been completed in multiple species, such as 162 *bHLH* genes in *Arabidopsis thaliana* [14], 192 *bHLH* genes in tobacco [27], 197 *bHLH* genes in pear [7]. We identified 162 *HhbHLH* genes in this study using data from the *H. hamabo* genome. Based on their chromosomal location, these genes were named *HhbHLH1–HhbHLH162*. The conserved domains of the *H. hamabo* bHLH protein were evaluated by multiple sequence alignment. According to the conserved domains, the 162 HhbHLH proteins could be divided into 19 subgroups. The specific number of subgroup classifications of the current plant bHLH family does not have a unified standard but is usually considered to be between 15–32 subgroups [7,14,28]. The number of genes in the different subgroups of the *H. hamabo* bHLH family varies from 1 to 20. Compared with *Arabidopsis*, the *bHLH* genes in *H. hamabo* are different among the subgroups, such as VIIa+b, Ia, and IVa, by the degree of expansion or contraction.

The type and composition of motifs can determine the function of the protein, and the evolutionary relationship between HhbHLH proteins can also be determined by analyzing their conservative motifs. In this study, the online MEME program was used to analyze the motifs of 162 HhbHLH proteins, and a total of 20 motif structures were obtained. The composition patterns of these motifs are almost the same as the results of phylogenetic analysis, the composition is similar in the same subfamily. At the same time, we analyzed the structure of all *HhbHLH* genes and found that *bHLH* members in the same group had similar gene structures. These findings further established the validity of the *HhbHLH*s classification.

Gene duplication events are very common in the evolution of plants, and they frequently play a critical role in the growth of gene families [29]. The 162 *HhbHLH* genes were divided into three types of repetitive events, of which 150 (92.59%) genes were assigned to WGD/segmental events, and 11 (0.067%) genes belonged to the dispersed type, which indicates that WGD/segmental events occur in *H. hamabo*. In addition, with synteny analysis, it was found that 60 and 26 *HhbHLH* genes have a syntenic relationship with *Arabidopsis* and *poplar* bHLH families, respectively. Some of the *HhbHLH* genes are connected to at least three syntenic gene pairs (especially in *Arabidopsis thaliana*), such as *HhbHLH11* and *HhbHLH141*, these genes may have played a key role in the *bHLH* gene family.

Substantial evidence proved that the *bHLH* gene family plays a critical role in plants, especially in plant response to drought and salt stress. For example, overexpression of the *MdSAT1* gene in apples increases the resistance of apple callus and transgenic *Arabidopsis* to drought and salt stresses [30], and the *CsbHLH041* gene in cucumber enhances the tolerance of transgenic cucumber seedlings to salt and ABA [15]. In this study, we analyzed the transcriptome data of *H. hamabo* under drought and salt stresses and revealed the expression pattern of the *HhbHLH* genes under these stresses. Except for a few genes, the results indicated that the expression of the majority of *HhbHLHs* changed significantly. After that, 15 highly expressed genes under drought and salt stress were screened and further analyzed by RT-qPCR. The results showed that the gene expression was consistent with the transcriptome data at the same time, which proved the reliability of the transcriptome data. In addition, three time points (1 h, 2 h, 12 h) were added to the analysis of the expression profiles of these genes under drought and salt stress. These results effectively supplemented the transcriptome data and provided a basis for the exploration of functional genes in the future. After that, we analyzed the expression of these 15 genes under heat, cold, ABA, and SA treatment, and screened a *HhbHLH2* gene with significant differences under various stresses for further study. Previous studies have found that the *Arabidopsis* IX subfamily *AtbHLH122* gene can be expressed in large quantities under drought stress, and compared with wild plants, the plants with the *AtbHLH122* gene overexpressed have stronger tolerance to drought stress [26]. In this study, the overexpression of Arabidopsis thaliana confirmed that *HhbhLh2* can improve the drought and salt stress tolerance of transgenic plants. In conclusion, the present study represents the first analysis of the *bHLH* gene family of the semi-mangrove plant *H. hamabo*, and the isolated identification of these transcription factors may help elucidate the molecular genetic basis of *H. hamabo* in response to drought and salt stress as well as reveal the importance of *bHLH* genes in response to abiotic stress.

In conclusion, this is the first comprehensive and systematic analysis of bHLH transcription factors in the semi-mangrove plant *H. hamabo*, and isolation and identification of these transcription factors help clarify the molecular genetic basis of *H. Hamabo* response to drought and salt stress and reveal the importance of *bHLH* genes in response to abiotic stress. The expression analysis of *HhbHLHs* genes also lays a strong foundation for future studies on the regulatory functions of bHLH proteins during abiotic stress response in *H. Hamabo* and will be able to further help understand this gene family in other plants. In addition, the functional identification of the *HhbHLH2* gene also provides a basis for further study on the regulatory network of *H. Hamabo* in response to drought and salt stress in the future. 

In conclusion, the results of this study reveal the importance of the *bHLH* gene in response to abiotic stress. This work may help to clarify the functions of *bHLH* family genes in protein interaction, signal pathway regulation, and defense response under different stress conditions, and provide basic resources for the future study of semi-mangrove plant tolerance to drought and salt stress. 

## 4. Materials and Methods

### 4.1. Gene Identification

The Pfam protein family database was queried for the HMM file corresponding to the bHLH domain (PF00010) (http://pfam.sanger.ac.uk/, accessed on 16 March 2021) [31]. The HMMER (v. 3.1) software was used to search for the *bHLH* gene in the *H. hamabo* genome database using default parameters and a cutoff value of 0.001 [32]. Some genes do not contain the bHLH domain and have no obvious structural features to be deleted artificially. The genome gene ID corresponding to *HhbHLH* is shown in Appendix A. The sequence length, molecular weight, and pI values for the bHLH protein were determined using the ExPasy website tools (http://web.expasy.org/protparam/, accessed on 16 March 2021).

### 4.2. Sequence Analysis

The default parameters of ClustalW (v. 2.1, C ++, USA) were used to align the protein sequence of the bHLH domain of the HhbHLH protein, and then GeneDoc (www.psc.edu/biomed/genedoc, accessed on 16 March 2021) software was used to manually adjust the amino acid sequence in the bHLH domain. MEME SUITE (v. 4.12.0, JAVA, Australia) software was used to analyze the conserved motifs in the HhbHLH protein. The following parameters were used: any number of repetitions; the maximum number was twenty, and the optimal width was between six and one hundred residues [33]. The *HhbHLH* gene structure was determined through the program GSDS (http://gsds.cbi.pku.edu.cn, accessed on 16 March 2021) [34]. Tbtools (v. 1.089, JAVA, China) were used to visualize the results [35].

### 4.3. Phylogenetic Analysis and Classification

According to the classification of *AtbHLH* in *Arabidopsis* and the comparison of the bHLH conserved domains of HhbHLH and AtbHLH proteins, the identified HhbHLHs were divided into different subgroups. The bHLH proteins of *Arabidopsis* were downloaded from the TAIR (https://www.arabidopsis.org/, accessed on 16 March 2021) A phylogenetic tree was constructed using the NJ method in MEGA 7.0 [36].

### 4.4. Chromosomal Distribution and Gene Duplication

The GFF file was used to extract the *HhbHLH* gene’s chromosome location information. The same procedure was used to analyze the synteny among the *HhbHLHs* in PGDD (http://chibba.agtec.uga.edu/duplication/, accessed on 16 March 2021). Primarily, local all-vs-all BLASTP searches were performed on identified *HhbHLH* genes. Following that, using the BLASTP result and gene location information as input files, MCScanX was used to determine syntenic gene pairs [37]. We used the MCScanX package downstream analysis tool (duplicate gene classifier) to identify tandem, proximal dispersed, and segmental/whole-genome duplications (WGD) of *HhbHLH* family genes. We analyzed the Ka and Ks values using the KaKs Calculator 2.0 [38]. To estimate the dates of segmental duplication events, the mean Ks was calculated for succeeding pairs of homologous genes within 100 Kb on all sides of the *HhbHLH* genes. Tbtools (v. 1.089, JAVA, China) were used to visualize the results.

### 4.5. Plant Materials and Treatments

*H. hamabo* seeds were collected from Nanjing’s Sun Yat-Sen Memorial Botanical Garden. The seeds were soaked in concentrated sulfuric acid for 10 min and then cultured in a greenhouse under aseptic conditions for 3 weeks (16 h/8 h light/dark; relative humidity 65%). Then, the seedlings with the same taproot length were transferred to a 50 mL centrifuge tube and cultured in half-strength Murashige and Skoog (1/2 MS) nutrient solution at pH 5.8 for 1 week before treatment. The seedlings were renewed every 2 days during the growth period. For drought or salt stress treatments, the two groups of seedlings were placed in 1/2 MS solution containing either 15% PEG6000 or 400 mM NaCl. For hormone treatment, *H. hamabo* seedlings were treated with 200 μM SA and 400 μM ABA. Plants were exposed to 4 °C and 42 °C in an incubator with the same photoperiod, respectively, for low- and high-temperature stressors. After stress treatments, leaves were sampled at 0, 1, 2, 6, 12, and 24 h. As controls, leaves sampled at 0 h were used. Each treatment was replicated biologically three times. After collecting the samples, they were promptly frozen in liquid nitrogen and then stored at –80 °C for future analysis.

### 4.6. RNA Extraction and Gene Expression Analysis 

A Plant Rneasy Mini Kit (Qiagen, Hilden, Germany) was used to extract total RNA according to the operating instructions. A spectrophotometer (NanoDrop2000, Thermo Scientific, Wilmington, DE, USA) was used to measure the RNA concentration at OD = 260 and 280 nm, and the threshold standard for 260/280 nm was 1.80–2.20. A PrimeScript^®^ RT kit (TaKaRa, Dalian, China) was used to synthesize cDNA, following the product manual, and 1 μg RNA was used each time to synthesize the first-strand cDNA. Genscript online design software (https://www.genscript.com/tools/pcr-primers-designer, accessed on 16 March 2021) was used to design primer pairs (Appendix A). The RT-qPCR was performed with StepOnePlus real-time PCR system (Applied Biosystems, Beijing, China). The qPCR parameters were 10 min at 95 °C; 15 s at 95 °C, 30 s at 60 °C, and 30 s at 72 °C, for 40 cycles; and the melting process was performed at 60–95 °C to generate a melting curve. The reaction mixture contains 10 μL of 2 × SYBR GreenMasterMix (Bimake, TX, USA), 1 μL of diluted cDNA, and 0.4 μL of forward and reverses primers (10 μM). *ACT* was used as an internal reference gene [39]. Three replicate analyses were performed for each reaction, and then the data was analyzed using the 2^−ΔΔCT^ method. Tbtools were used to draw a heat map of log_2_ (1 + 2^−ΔΔCT^) values. *HhbHLH* gene expression was analyzed according to the FPKM value of the transcriptome data of the *H. hamabo* under drought and salt stresses [40], and Tbtools was used to draw a heat map of the log_2_ (1+FPKM) value.

### 4.7. Generation of HhbHLH2 Transgenic Arabidopsis Plants

The ORF sequence of *HhbHLH2* was cloned into the *pCAMBIA1305* vector. Then *HhbHLH2-1305* was transformed into GV3101, and *Arabidopsis thaliana* Col-0 was transformed by the inflorescence infection method [41]. The T3 transgenic *Arabidopsis* was identified by hygromycin (150 mg/L) selection for subsequent experimental analysis. To treat *Arabidopsis thaliana* with salt stress and drought, the seeds of the transgenic *Arabidopsis* and Col-0 (WT) were sterilized and cultured on 1/2 MS medium containing 200 mM D-mannitol and 50 mM NaCl. The phenotypes were observed and the root length was measured after 7 days of cultivation at 22 °C.

## 5. Conclusions

Using the HMMER software, we identified 162 *HhbHLHs* in *H. hamabo*. Then the 162 *HhbHLH* genes were split into 19 subfamilies. Gene structure and motif analyses corroborated the phylogenetic analysis results. The *cis*-elements found in the promoters of the *HhbHLHs* were found to be associated with phytohormones and abiotic stresses. Transcriptomic profiles and RT-qPCR analysis revealed that *HhbHlH57*, *HhbHlH78*, *HhbHlH136*, and *HhbHlH78* were significantly up-regulated in response to salt and drought stress. In addition, the overexpression of *HhbHLH2* increased salt and drought stress tolerance in *Arabidopsis*.

## Figures and Tables

**Figure 1 ijms-22-08748-f001:**
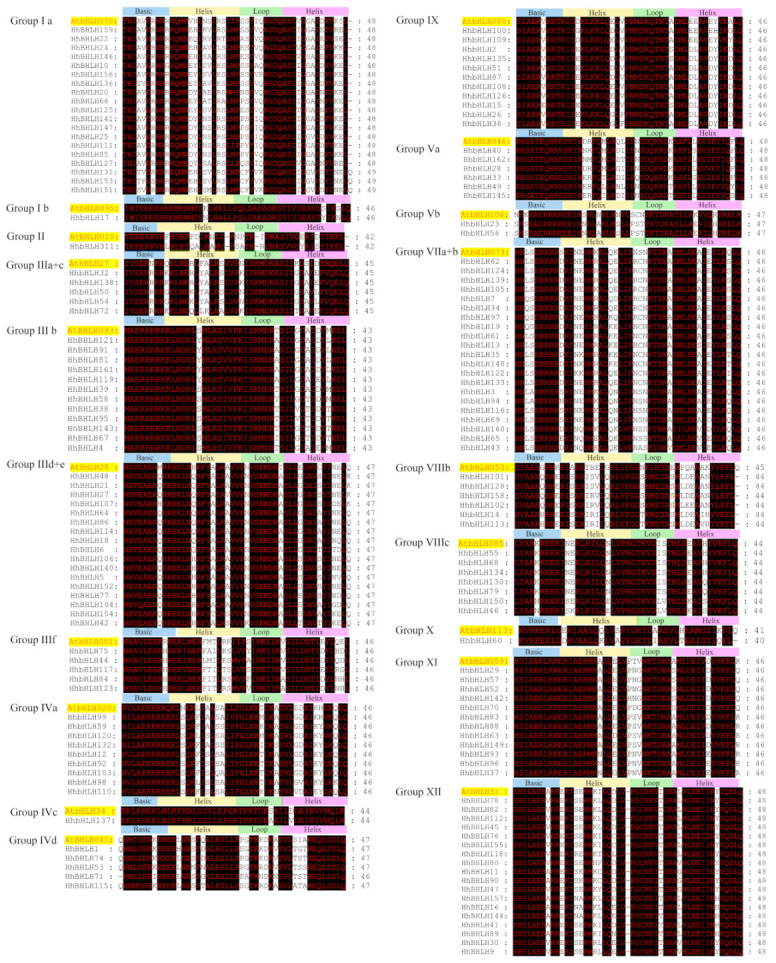
Alignment of the amino acid sequences of multiple HhbHLH and selected Atb HLH domains.

**Figure 2 ijms-22-08748-f002:**
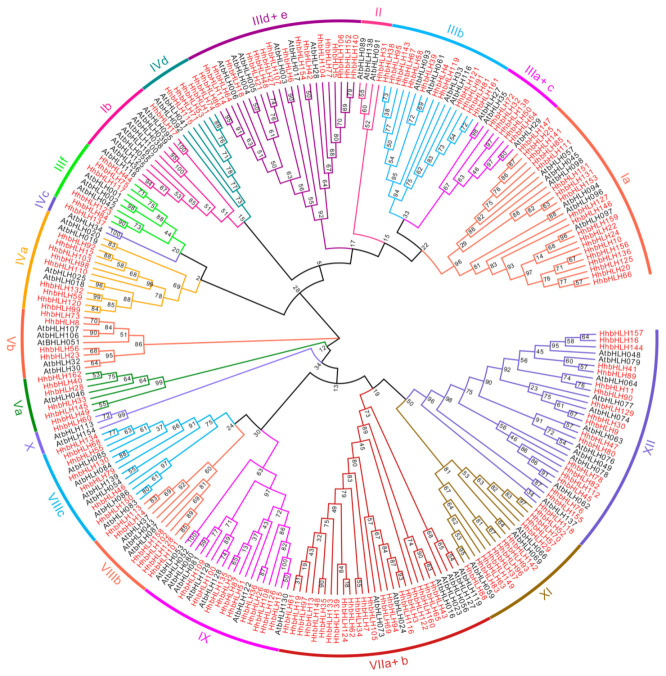
Unrooted phylogenetic tree of bHLH proteins from *Arabidopsis thaliana*, and *H. hamabo*. MEGA7 was used to construct the phylogenic tree from the sequences of the domain protein. Evolview was used to annotate and analyze the phylogenic tree. Various background colors show the HhbHLH protein group.

**Figure 3 ijms-22-08748-f003:**
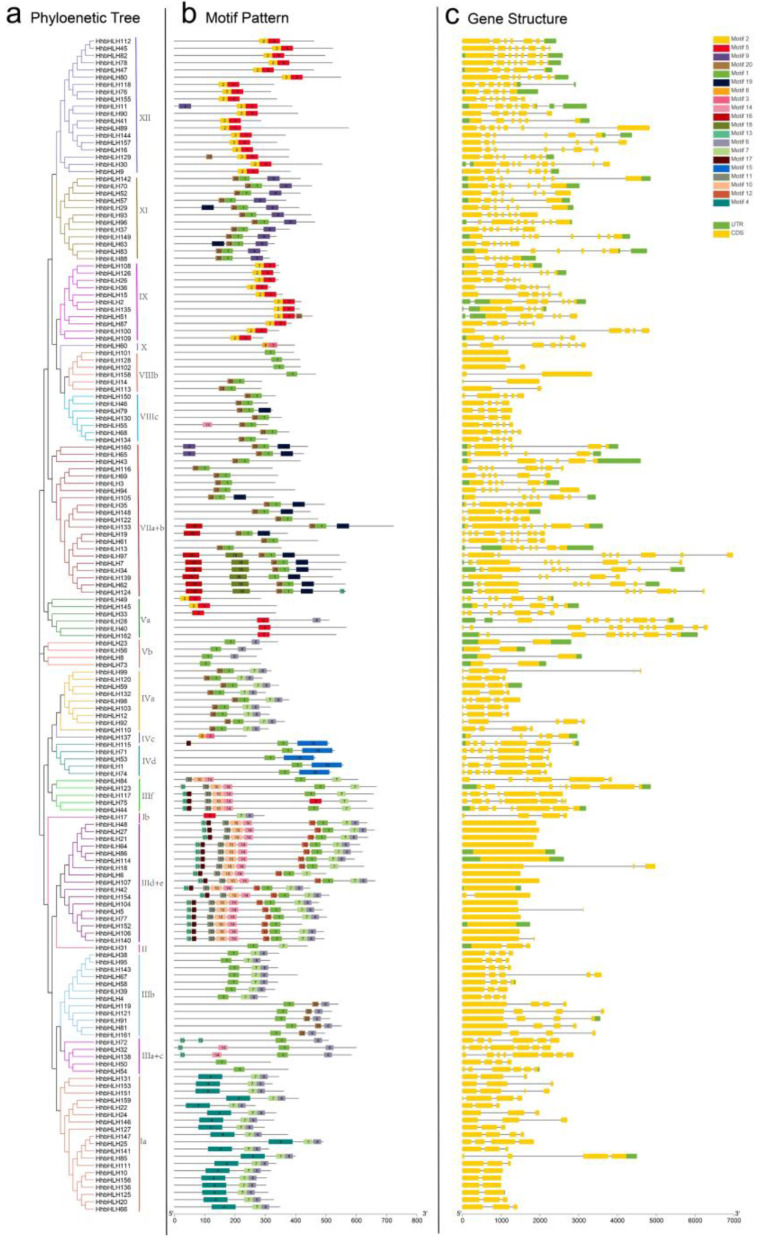
Phylogenetic relationships, gene structure, and architecture of conserved protein motifs in *bHLH* genes from *H. hamabo*. (**a**) The phylogenetic tree and the subfamily are shown in different colors. (**b**) The motif composition of *H. hamabo* bHLH proteins. (**c**) The exon-intron structure of *H. hamabo bHLH* genes.

**Figure 4 ijms-22-08748-f004:**
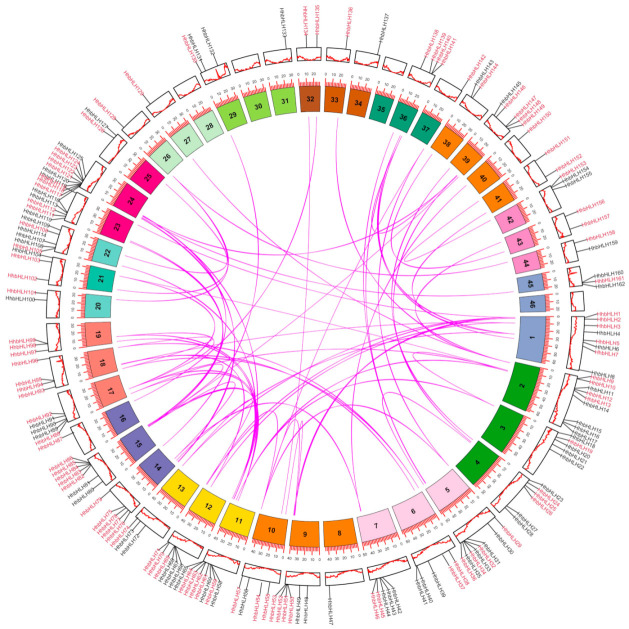
Distribution and collinearity of the *HhbHLHs*. Black lines along the circumference of the circle mark the positions of genes on chromosomes. The lines inside the circle indicate collinearity relationships among *HhbHLH* genes.

**Figure 5 ijms-22-08748-f005:**
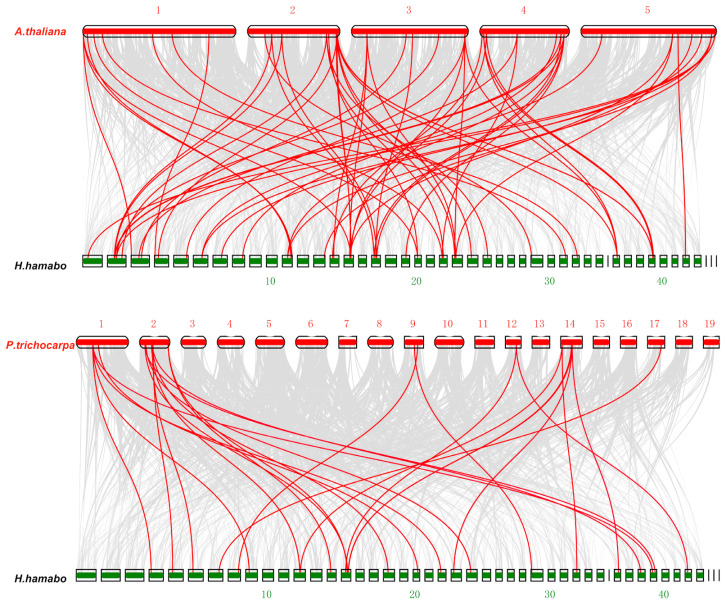
Synteny analysis maps of *H. hamabo* with *Arabidopsis* and *Populus*. Gray lines in the background indicate the collinear blocks within *H. hamabo* and other plant genomes, while the red lines highlight the syntenic *bHLH* gene pairs.

**Figure 6 ijms-22-08748-f006:**
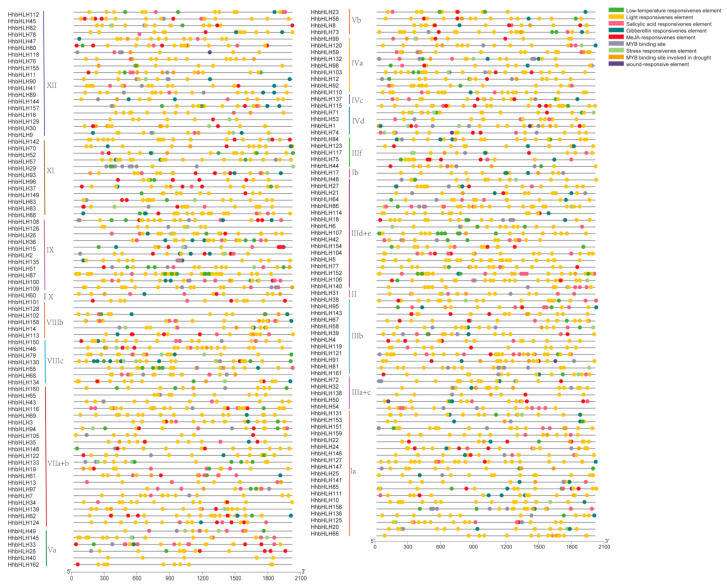
Analysis of the promoters of 162 *bHLH* genes in *H. hamabo* using *cis*-element analysis. Different hues denoted substances associated with plant hormones (salicylic acid, gibberellin, and methyl jasmonate) and stress resiliency (light, low temperature, wound-responsive, and drought inducibility).

**Figure 7 ijms-22-08748-f007:**
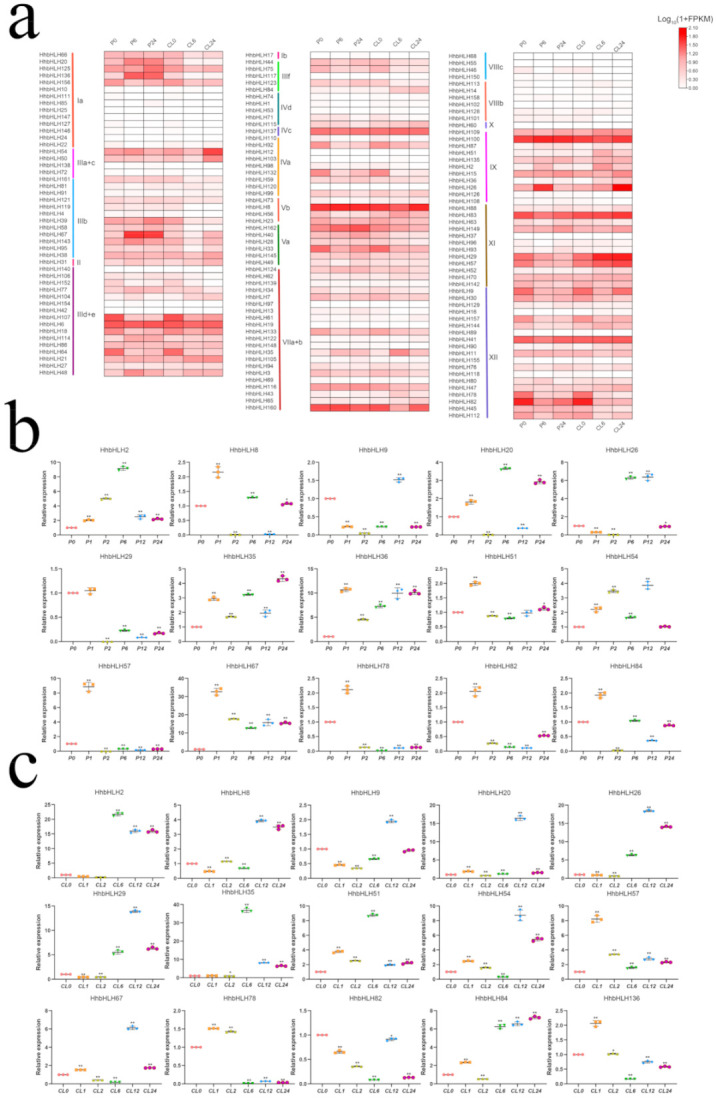
Expression profile of *HhbHLHs* under drought and salt stresses. (**a**) A heatmap shows the expression level of 162 *HhbHLH* genes with different subfamilies in salt and drought stress of *H. hamabo*. Expression differences are observed in different colors. The P0, P6, P24 represent 0 h, 6 h, and 24 h were treated with PEG; CL0, CL6, and CL24 represent 0 h, 6 h, and 24 h were treated with NaCl. A color change indicates a change in expression level; white indicates a lower level of expression, whereas red indicates a higher level of expression. (**b**) Relative expression of 15 *HhbHLH* genes chose at random with PEG treatment, P0, P1, P6, P12, and P24 represent 0 h, 1 h, 2 h, 6 h, 12 h, and 24 h were treated with PEG. (**c**) Relative expression of 15 *HhbHLH* genes chosen at random with NaCl treatment, CL0, CL1, CL6, CL12, and CL24 represent 0 h, 1 h, 2 h, 6 h, 12 h and 24 h were treated with NaCl. The *ACT* gene was used to normalize the data, and vertical bars represent standard deviation. Asterisks denote genes that are significantly up-or down-regulated in comparison to the untreated control (* *p* < 0.05, ** *p* < 0.01, Student’s *t*-test).

**Figure 8 ijms-22-08748-f008:**
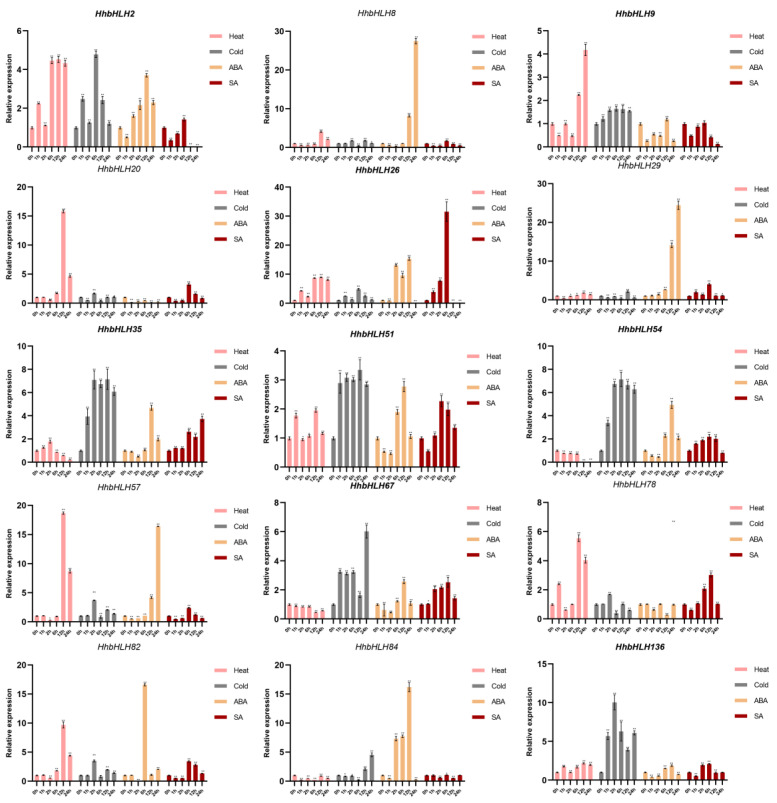
Expression profiles of 15 selected *HhbHLH* genes in response to various stress treatments. Different colors represent different stress treatments. Pink represents heat stress, gray represents cold stress, yellow represents ABA stress, and red represents SA stress. Asterisks denote genes that are significantly up- or down-regulated in comparison to the untreated control (* *p* < 0.05, ** *p* < 0.01, Student’s *t*-test).

**Figure 9 ijms-22-08748-f009:**
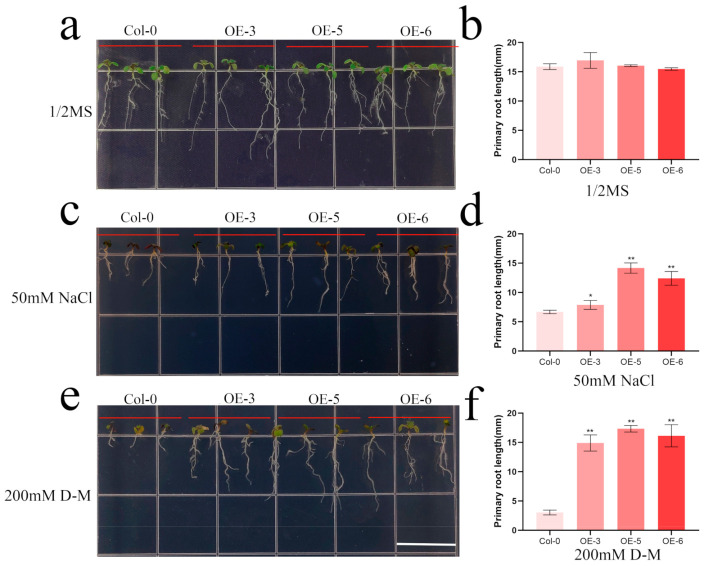
Effects of salt and drought stress on root length of wild-type (col-0) and *Arabidopsis* overexpressing *HhbHLH2* (OE-3, OE-5, and OE-6). (**a**) Three-day-old seedlings were transferred to MS medium for 7 days. (**b**) Root length statistics in MS medium. (**c**) Three-day-old seedlings were transferred to MS medium in 50 mM NaCl for seven days. (**d**) Root length statistics in 50 mM NaCl for seven days. (**e**) Three-day-old seedlings were transferred to MS medium in 200 mM D-M for 7 days. (**f**) Root length statistics in MS medium 200 mM D-M for 7 days. Asterisks indicate the corresponding gene significantly up- or down-regulated compared with the untreated control (* *p* < 0.05, ** *p* < 0.01, Student’s *t*-test).

**Table 1 ijms-22-08748-t001:** Numbers of *bHLH* genes from different origins in *H. hamabo*.

Duplication Type	Singleton	Dispersed	Proximal	Tandem	WGD/Segmental
No. of *bHLH* genes from different origins (percentage)	0 (0.00%)	11(6.79%)	1 (1.62%)	0(0.00%)	150 (92.59%)

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
