# Peer review of "Genome-wide Analysis of Basic Helix-Loop-Helix Family Genes and Expression Analysis in Response to Drought and Salt Stresses in Hibiscus hamabo Sieb. et Zucc"

_ijms, 2021, doi:10.3390/ijms22168748_

Round 1

Reviewer 1 Report

The authors have put together a comprehensive review of bHLH transcription factors in the species H. hamabo and their connection to drought stress. This is of potential interest to those who study H. hamabo, or drought in other species, and the science is of good quality. I have identifed a few places in the manuscript where language needs to be clarified or detail added. And I suggest the authors consider changing the format of Figure 8 in particular to improve its read-ability. Please see my attached word doc for all comments.

Author Response

Thank you very much for useful comments of our manuscript! We have made careful revisions, and the detailed corrections are listed below point by point. 

Please let me know if you have any question!

Thank you very much for your attention!

With best regards,

Yours sincerely,

Chunsun Gu

Reviewer 1 Comments

Open Review

English language and style

( ) Extensive editing of English language and style required
( ) Moderate English changes required
(x) English language and style are fine/minor spell check required
( ) I don't feel qualified to judge about the English language and style

Yes

Can be improved

Must be improved

Not applicable

Does the introduction provide sufficient background and include all relevant references?

(x)

( )

( )

( )

Is the research design appropriate?

(x)

( )

( )

( )

Are the methods adequately described?

(x)

( )

( )

( )

Are the results clearly presented?

( )

( )

(x)

( )

Are the conclusions supported by the results?

(x)

( )

( )

( )

Comments and Suggestions for Authors

The authors have put together a comprehensive review of bHLH transcription factors in the species H. hamabo and their connection to drought stress. This is of potential interest to those who study H. hamabo, or drought in other species, and the science is of good quality. I have identified a few places in the manuscript where language needs to be clarified or detail added. And I suggest the authors consider changing the format of Figure 8 in particular to improve its readability. Please see my attached word doc for all comments.

Point 1: Line 41: Replace the word “predominantly” with “important for”. bHLH TFs are involved in all sorts of processes beyond defense responses, and as written this statement is misleading.

Response 1: Thank you! We have replaced the word “predominantly” with “important for” ( lines 41 ).

Point 2: Line 79-80: Correct wording to “There were 167 putative bHLH proteins discovered.”

Response 2: Thank you! We have modified this sentence ( lines 79-80 ).

Point 3:Line 186 -187: I don’t understand this sentence “This phenomenon suggests that H. hamabo may be related to the phylogeny between Arabidopsis.” Is it missing an object at the end? The connection to the synteny with Arabidopsis and H. hamabo described earlier in the paragraph is not clear at all.

Response 3: Thank you! I am sorry that I did not state this issue clearly. We have modified this sentence as below “This phenomenon suggested that an evolutionary relationship may be existing between the bHLH gene family of H. hamabo and that of Arabidopsis ( lines 186-187 ).

Point 4:Figure 6. Not high enough resolution to read the legend text on the PDF.

Line 207. Briefly clarify how drought stress and/or salt stress were induced before RNA extraction.

Response 4: Thank you! We have re-uploaded the high-resolution images ( Fig. 1-9 ) and relative details were added in the revised manuscript (lines 208-209 ).

Point 5:Line 229: Remove hyphen from “three-time”, and list here what time points were added and why they were chosen.

Response 5:Thank you! We have modified it ( lines 227-232 ).

Point 6:Line 239 – 241. Unless this topic is explored in the discussion section, a sentence explaining the most likely hypothesis for when there are differences between the transcriptomic data and the qRT-PCR would be a good build here. Is it the addition of more time points? Is it the difference between NaCl and PEG treatments?

Response 6:Thank you! We've explained this part ( lines 240-245 ).

Point 7:Line 248, 250. Both panels (b) and (c) say that they are PEG treatment, but the lettering (P versus C) and the later text in the panel would indicate that these are two different types of treatments, PEG and NaCl. Please confirm which data type we are looking at clarify both in the figure legend, and in the text in the paragraph between lines 226-241.

Response 7:Thank you! I am sorry I made a mistake here, and we have corrected it in the revised version ( lines 256-257 )

Point 8:Line 277. Should T1 correlate to 1h? It would be simper just to list these as 0h, 1h, 2h, etc. than using the “T#” nomenclature.

Response 8:Thank you! We have modified it. (Fig.8)

Point 9:Figure 8. This is a lot of plots to look at. Consider using different colors to indicate the different treatments. I also think it would be more useful to reader to display the data so that the same gene profiles could be compared across treatments. Currently the chart in the top right of each panel corresponds to the same gene, but if each were a different color and they were stacked vertically, then you could compare both across time (the x-axis) and between treatments (the vertical column of charts). (If I am incorrect and the genes do not correspond between panels, forgive me, I cannot read the labels on the PDF.)

Response 9:Thank you! We have redrawn Fig.8 in the revised version(Fig.8, lines 283-284).

Point 10:Line 287. The word “transferred” here is imprecise. Do you mean introduced by stable transformation via agrobacterium? More methodological detail is required in the text.

Response 10: Thank you! We've added more details in the revised version(lines 293-296).

Point 11:Line 298. Would like to see a sentence here comparing these results to those of the bHLH122 mutant in Arabidopsis under salt stress. How similar is this phenotype?

Response 11: Thank you! We've added more details in the revised version (lines 295-304).

Point 12:Line 342. Plant tolerance of…what?

Response 12: Thank you! The sentence was revised as “Substantial evidences proved that the bHLH gene family plays a critical role in plants, especially in plant response to drought and salt stress.” ( lines 352-353 ).

Point 13:Discussion section: This discussion is mostly a summary of the results as already stated in the results section, without significant building. Please condense the existing summary and include more future plans for what functional studies are proposed and how those will add to new discoveries about drought tolerance.

Response 13: Thank you! We have modified the discussion section according to your advise ( lines 356-392 ).

Point 14:Line 394-395. If genes were manually deleted that did not contain the bHLH domain, how did they end up in the PFam output? Were these genes compared with anything else in the phylogeny? Is it possible they represent a new/different class of bHLH that do not contain the canonical domain? If there was a good rationale for why those genes were deleted/excluded it should be explained here.

Response 14: Thank you! We've explained it in the revised version ( lines 398-399 ).

Reviewer 2 Report

The manuscript is at a good level and on an interesting topic. It contains only minor bugs that need to be fixed:

  • unify the writing of genes, gene families, proteins according to standards, i.e. genes / gene families in italics.
  • unify the format of journal names, including respect for uppercase and lowercase letters in the names.
  • supplementes - wrong number format (decimal point)
  • line 149 - discussion in results?
  • some figures very small and poor resolution, reduces the level of handwriting and the ability to verify the achieved and discussed results.

I recommend taking the manuscript after minor revision.

Author Response

Thank you very much for useful comments of our manuscript! We have made careful revisions, and the detailed corrections are listed below point by point. 

Please let me know if you have any question!

Thank you very much for your attention!

With best regards,

Yours sincerely,

Chunsun Gu

Response to Reviewer 2 Comments

Open Review

English language and style

( ) Extensive editing of English language and style required
( ) Moderate English changes required
(x) English language and style are fine/minor spell check required
( ) I don't feel qualified to judge about the English language and style

Yes

Can be improved

Must be improved

Not applicable

Does the introduction provide sufficient background and include all relevant references?

(x)

( )

( )

( )

Is the research design appropriate?

(x)

( )

( )

( )

Are the methods adequately described?

(x)

( )

( )

( )

Are the results clearly presented?

(x)

( )

( )

( )

Are the conclusions supported by the results?

(x)

( )

( )

( )

Comments and Suggestions for Authors

The manuscript is at a good level and on an interesting topic. It contains only minor bugs that need to be fixed:

  • unify the writing of genes, gene families, proteins according to standards, i.e. genes / gene families in italics.
  • unify the format of journal names, including respect for uppercase and lowercase letters in the names.
  • supplementes - wrong number format (decimal point)
  • line 149 - discussion in results?
  • some figures very small and poor resolution, reduces the level of handwriting and the ability to verify the achieved and discussed results.

I recommend taking the manuscript after minor revision.

Point 1: unify the writing of genes, gene families, proteins according to standards, i.e. genes / gene families in italics.

Response 1: Thank you! We have modified it in the revised version (For example, lines 13, 19, 52, 71).

Point 2: unify the format of journal names, including respect for uppercase and lowercase letters in the names.

Response 2: Thank you! We have modified it in the revised version (lines 503-601).

Point 3: supplementes - wrong number format (decimal point).

Response 3: Thank you! We have modified it in the revised version (Table S2, S4, S6)

Point 4: line 149 - discussion in results?

Response 4: Thank you! We have modified the discussion section according to your advise ( lines 354-390 ).

Point 5:• some figures very small and poor resolution, reduces the level of handwriting and the ability to verify the achieved and discussed results.

Response 5: Thank you! We have re-uploaded the high-resolution figures ( Fig. 1-9 ).
